# Vacancy-defect modulated pathway of photoreduction of $CO_2$ on single atomically thin $AgInP_2S_6$ sheets into olefiant gas

Wa Gao[1,9], Shi Li[2,9], Huichao He [3], Xiaoning Li [4], Zhenxiang Cheng [4], Yong Yang [5], Jinlan Wang [2✉], Qing Shen[6], Xiaoyong Wang [1], Yujie Xiong[7✉], Yong Zhou [1,8✉] & Zhigang Zou[1,8]

Artificial photosynthesis, light-driving $CO_2$ conversion into hydrocarbon fuels, is a promising strategy to synchronously overcome global warming and energy-supply issues. The quaternary $AgInP_2S_6$ atomic layer with the thickness of ~ 0.70 nm were successfully synthesized through facile ultrasonic exfoliation of the corresponding bulk crystal. The sulfur defect engineering on this atomic layer through a $H_2O_2$ etching treatment can excitingly change the $CO_2$ photoreduction reaction pathway to steer dominant generation of ethene with the yield-based selectivity reaching ~73% and the electron-based selectivity as high as ~89%. Both DFT calculation and *in-situ* FTIR spectra demonstrate that as the introduction of S vacancies in $AgInP_2S_6$ causes the charge accumulation on the Ag atoms near the S vacancies, the exposed Ag sites can thus effectively capture the forming *CO molecules. It makes the catalyst surface enrich with key reaction intermediates to lower the C-C binding coupling barrier, which facilitates the production of ethene.

[1] Key Laboratory of Modern Acoustics (MOE), Institute of Acoustics, School of Physics, Jiangsu Key Laboratory of Nanotechnology, Eco-materials and Renewable Energy Research Center (ERERC), National Laboratory of Solid State Microstructures, Collaborative Innovation Center of Advanced Microstructures, Nanjing University, Nanjing, China. [2] School of Physics, Southeast University, Nanjing, China. [3] State Key Laboratory of Environmental Friendly Energy Materials, Southwest University of Science and Technology, Mianyang, China. [4] Institute of Superconducting & Electronic Materials, Innovation Campus, University of Wollongong, Squires Way, North Wollongong, NSW, Australia. [5] Key Laboratory of Soft Chemistry and Functional Materials (MOE), Nanjing University of Science and Technology, Nanjing, China. [6] University of Electrocommunication, Grad Sch Informatics and Engineering, Chofu, Tokyo, Japan. [7] Hefei National Laboratory for Physical Sciences at the Microscale, Collaborative Innovation Center of Chemistry for Energy Materials (iChEM), School of Chemistry and Materials Science, University of Science and Technology of China, Hefei, Anhui, China. [8] School of Science and Engineering, The Chinese University of Hongkong (Shenzhen), Shenzhen, Guangdong, China. [9] These authors contributed equally: Wa Gao, Shi Li. ✉email: jlwang@seu.edu.cn; yjxiong@ustc.edu.cn; zhouyong1999@nju.edu.cn

Photocatalytic conversion of $CO_2$ with $H_2O$ into solar fuels would be like killing two birds with one stone in terms of saving supplying energy and environment, which occurs mostly on the surfaces of semiconductors through complicated processes involving multi-electrons/protons transfer reactions[1]. Photo-driving $CO_2$ hydrogenation into $C_1$ species have been well achieved in the recent decade[2], and our group has exploited a series of promising photocatalysts to converse $CO_2$ to selectively form specific hydrocarbons, such as $Zn_2GeO_4$ ultrathin nanoribbons for $CH_4$[3], atomically thin $InVO_4$ nanosheets for CO[4], and $TiO_2$-graphene hybrid nanosheets for $C_2H_6$[5] and so on. However, the controlled C–C coupling to produce high-value $C_2$ or $C_{2+}$ products still remains a great challenge. Olefiant gas (ethylene, $C_2H_4$) is a chemical source of particular importance due to its high demand in the chemical industry. $C_2H_4$ is usually derived from steam cracking of naphtha under harsh production conditions (800–900 °C). It is definitely desirable for the realization of $C_2H_4$ synthesis through mild and environmentally benign pathways[6].

Transition metal thio/selenophosphates (TPS) is a broad class of van der Waals layered structures with two sulfur or selenium layers sandwiching a layer of metal ions and $P_2$ pairs and general compositions of $M_4[P_2X_6]^{4-}$, $[M^{2+}]_2[P_2X_6]^{4-}$, and $M^{1+}M^{3+}[P_2X_6]^{4-}$, where $M^{1+}$ = Cu, Ag; $M^{3+}$ = Cr, V, Al, Ga, etc. X = S, Se[7]. Those quaternary compounds exhibit mixed electron–ionic conductivity, promising optical and thermoelectric properties[8]. $AgInP_2S_6$ is a typical TPS with a rhombohedral structure and contains a sulfur framework with the octahedral voids filled by Ag, In, and P–P triangular patterns. Each $AgInP_2S_6$ monolayer consists of the $[P_2S_6]$ anionic complex and two metallic cations (Ag and In) located at the center of sulfur near-octahedral polyhedrons connected one with the other by edges. Semiconducting $AgInP_2S_6$ crystal possesses an appropriate bandgap structure ($E_g$ = ~2.4 eV), which is favored for visible light absorption[9]. The low value of the effective mass of electrons and the high value of the effective mass of holes facilitate accelerating the mobility dynamics of photogenerated electrons onto the surface prior to holes[10], which may enhance local electron density, benefiting the photo-driving reduction reaction. The centrosymmetry structure of $AgInP_2S_6$ also enables the photo-excited electrons to distribute on the surface of the layer crystal uniformly[11], which may remarkably reduce the energy barrier for catalytic molecule activation, alter the catalytic reduction pathway, and enhance yield and enrich species of products.

An atomically thin 2D structure is an ideal platform to provide atomic-level insights into the structure-activity relationship[12]. Firstly, the ultrathin structure allows the photo-generated carriers to easily transfer from the interior to the surface with shortened charge transfer distance, decreasing the bulk recombination. Secondly, large surface exposure renders rich catalytic active sites. Thirdly, transparency resulting from ultrathin thickness helps for light absorption. The creation of vacancy defects in the ultrathin structure can also additionally enrich the reaction intermediates, resulting in low-coordinated atoms on the surface of the catalyst, which are known to facilitate to the generation of multi-carbon species from $CO_2$ photoreduction[13,14].

Herein, we report the synthesis of the $AgInP_2S_6$ single atomic layer (abbreviated as SAL) of ~0.70 nm in thickness through a facile probe sonication exfoliation of the corresponding bulk crystal (abbreviated as BC). The sulfur vacancy (abbreviated as $V_S$) defects were introduced in the resulting SAL through an etching process with $H_2O_2$ solution (abbreviated as $V_S$-SAL), which was prospectively utilized for photocatalytic reduction of $CO_2$ in the presence of water vapor. While BC and SAL dominantly produce CO, the implemented defect engineering changes the reaction pathway of the $CO_2$ photoreduction on $V_S$-SAL,

which allows steering $CO_2$ conversion into $C_2H_4$ with the yield-based selectivity reaching ~73% and the electron-based selectivity as high as ~89%, and the quantum yield of 0.51% at a wavelength of 415 nm. Both DFT calculation and in situ FTIR spectra demonstrate that the key step for the CO production on BC and SAL follows a conventional hydrogenation process of $CO_2$ to form *COOH, which further couples a proton/electron pair to generate *CO. *CO easily liberates from the defect-free $AgInP_2S_6$ surface with low absorption energy to become free CO gas. In contrast, the introduction of $V_S$ in $AgInP_2S_6$ causes the charge accumulation on the Ag atoms near $V_S$. Thus, the exposed Ag site in $V_S$-SAL can effectively capture the forming *CO, making the catalyst surface enrich with key reaction intermediates to promote C–C coupling into $C_2$ species with the low binding energy barrier. This work may provide fresh insights into the design of an atomically thin photocatalyst framework for $CO_2$ reduction and establish an ideal platform for reaffirming the versatility of defect engineering in tuning catalytic activity and selectivity.

## Results

**Structure characterization of the $AgInP_2S_6$ related samples**. BC was synthesized through PVT in a two-zone furnace, which displays bright yellowish-brown color (Supplementary Fig. 1a). The SAL was produced through mechanical exfoliation in ethyl alcohol solution through a probe sonication technique, which can transfer high energy into layered materials and weaken the Van der Waals forces between adjacent layers, resulting in effective delamination. The well-defined Tyndall effect of the resulting transparent solution of SAL indicates high monodispersity of the ultrathin sheets (Supplementary Fig. 1b). Etching of SAL with $H_2O_2$ solutions allows to deliberately create $V_S$ on the surface of SAL[15].

The powder X-ray diffraction (XRD) pattern of BC and SAL agrees with the simulated one from the crystal structure of ICSD 202185 well with the $P_{\bar{3}1c}$ space group (Supplementary Fig. 2)[12], and no impurity peaks were detected. The stronger SAL peak intensity ratio of (002) to (112) relative to BC indicates that the exfoliation of $AgInP_2S_6$ occurs along [001] direction. The field emission scanning electron microscopy (FE-SEM) image shows that BC displays an angular shape with an apparent laminar structure (Supplementary Fig. 3a, b). The energy dispersive spectroscopy (EDS) spectra demonstrate the uniform spatial distribution of Ag, In, P, and S (Supplementary Fig. 3c–f). The TEM image of exfoliated SAL displays light contrast of the extremely thin 2D structure (Fig. 1a). A magnified transmission electron microscopy (TEM) image of a vertically standing sheet shows the single layer with a thickness of ~0.71 nm (Fig. 1b). A typical edge-curling sheet as marked with an arrow also particularly shows the thickness of ~0.72 nm of SAL (Fig. 1b), well in agreement with the $AgInP_2S_6$ monolayer along [002] orientation [$d_{(002)}$ = 6.68 Å]. The corresponding atomic force microscopy (AFM) image of SAL also confirms ~0.66–0.73 nm range in thickness (Fig. 1c and Supplementary Fig. 4 for more images), demonstrating the single-atom layer feature. A high-resolution TEM (HRTEM) image of SAL reveals that the interplanar $d$-spacing between the well-defined lattice fringes were examined 0.54 nm, which can be indexed to (010) (Fig. 1d). The selected area electron diffraction shows an ordered array of spots recorded from [001] zone axis (Fig. 1d, inset), confirming that SAL is of single crystallinity and preferentially enclosed by {002} top and bottom surfaces. The crystalline model of SAL from top and side views was schematically illuminated in Fig. 1f. With $H_2O_2$ solution treatment for optimized 10 s, the sulfur atoms, which locate outermost in SAL, can be partially etched away from the surface to form $V_S$. The generation of $V_S$ was

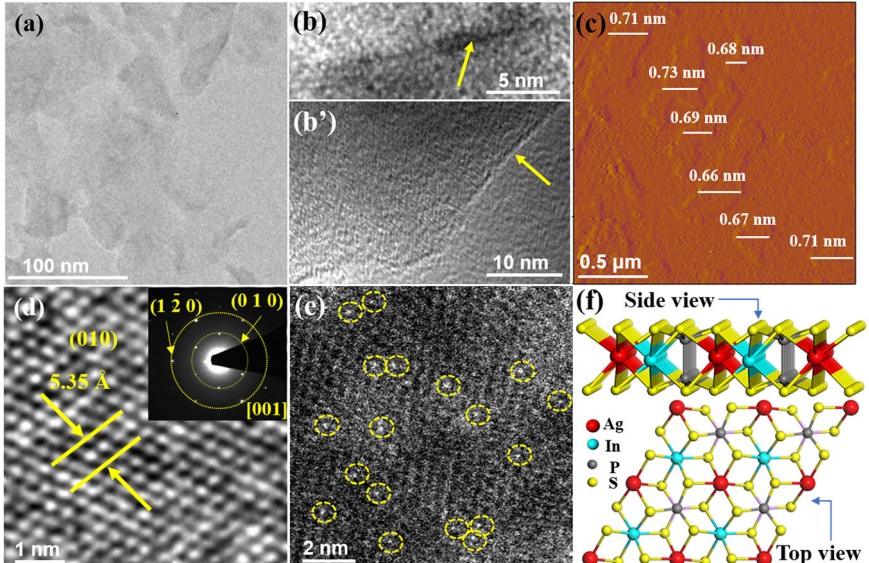

**Fig. 1 Morphological structure characterization of the fabricated SAL and V$_S$-SAL$_{10}$.** TEM images of **a** SAL, **b** vertically standing, and **b'** laying single piece SAL, **c** AFM image of SAL showing an average thickness of ~0.69 nm. **d** HRTEM image and the EDS. **e** HAADF-STEM image of V$_S$-SAL$_{10}$, in which the atomically dispersed V$_S$ are highlighted with the yellow circles. **f** The crystalline models of SAL from top and side views.

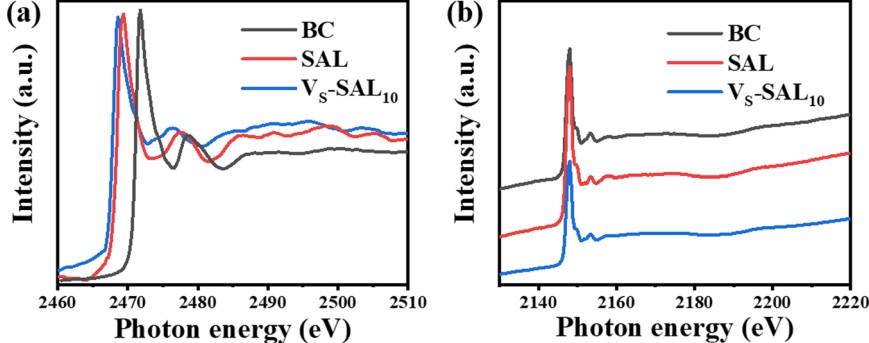

**Fig. 2 XANES spectra of BC, SAL, and Vs-SAL10. a** S and **b** P K-edge XANES spectra of BC, SAL, and Vs-SAL10.

confirmed with the electron paramagnetic resonance (EPR) spectra (Supplementary Fig. 5). The Raman spectra show that the peak intensity of both S–P–P and P–S–P for Vs-SAL were lowered, compared with those of SAL (Supplementary Fig. 6), which additionally verifies the detected defect sites can be assigned to V$_S$[16], rather than the open of S–M (metal) bond or the possible insertion of O atoms.

No obvious difference of the XRD patterns between SAL and Vs-SAL demonstrates no crystal structure change of the SAL before and after H$_2$O$_2$ etching treatment (Supplementary Fig. 2). The TEM image also shows that the resulting V$_S$-SAL$_{10}$ displays no morphology change in ultrathin structure (Supplementary Fig. 7). The corresponding EDS reveals that Ag, In, and P contents were nearly stoichiometric 1:1:2 of AgInP$_2$S$_6$, expect S element less than the stoichiometric ratio (Supplementary Fig. 8). It indicates that H$_2$O$_2$ treatment mainly leads to V$_S$, and has no etching effect on other moieties, which was also verified with the following XPS and the X-ray absorption near edge structure (XANES) spectra. The atomic resolution, aberration-corrected high-angle annular dark-field scanning TEM (HAADF-STEM) clearly reveals that a considerable number of V$_S$ were confined in the sheet (Fig. 1e), in contrast to the few sporadic ones in SAL (Supplementary Fig. 9).

Full XPS spectra demonstrate the presence of Ag, In, P, and S (Supplementary Fig. 10a). The high-resolution S $2p$ spectrum of

BC shows the S $2p$ peak falling between 162 and 164 eV (Supplementary Fig. 10b), revealing the −2 oxidation state of S. The S $2p$ peaks of SAL show dramatic low binding energy shift, compared with BC, and V$_S$-SAL$_{10}$ possesses further low-energy shift. The former shift may originate from exfoliation-resulting monolayerization[17] and the latter from V$_S$[15]. As the decrease of binding energy indicates the enhanced electron screening effect due to the increase of the electron concentration[15,18], it implies that the electron density around the S sites increases in the sequence of BC, SAL, and V$_S$-SAL$_{10}$. It reveals that the residual S atoms exist in an electron oversaturated form and possess high electron density. No obvious change of binding energy of P elements was observed (Supplementary Fig. 10c), further demonstrating that the mechanical exfoliation and chemical etching only damage sulfur atoms and have little effect on P moiety. Weak O$1s$ XPS peaks were observed for both SAL and Vs-SAL$_{10}$ (Supplementary Fig. 10d), which more likely originate from absorbed components from the ambiance. The almost same intensity and location of O$1s$ peak indicate no apparent oxidation change before and after H$_2$O$_2$ treatment. The pre-edge characteristic of the XANES spectra of the S K-edges of three AgInP$_2$S$_6$ was shown in Fig. 2a, which could be fitted with components of a spin−orbit split. The spectra indicate the existence of main transitions energies between 2460 and 2500 eV, which originates from the excitation of an electron from a 1S

inner orbital to a higher-energy orbital as a result of interaction with an X-ray. In comparison with BC, SAL shows a shift for S K-edge peaks to the lower energy side. This can be explained by the fact that the core electrons of S become more loosely bound after mechanical exfoliation due to the increased screening of the nuclear charge. Through $V_S$ engineering, the S K-edge of $V_S$-$SAL_{10}$ can have a further small move to the lower energy side (Fig. 2a). Moreover, the K-edge peak of P between 2100 to 2250 eV exhibits almost no differences among BC, SAL, and $V_S$-$SAL_{10}$ (Fig. 2b), which is in good agreement with the above-mentioned XPS results.

The UV–vis diffuse reflectance spectra show that the bandgap of SAL was determined 2.66 eV, a little larger than that of BC (2.31 eV) (Supplementary Fig. 11), exhibiting a strong quantum size effect in the lateral direction. $V_S$-$SAL_{10}$ displays a slightly narrowed bandgap (2.57 eV) with respect to SAL. It derives from that introduction of $V_S$ may tailor the electronic structure of SAL through generating impurity states near the conduction band (CB) edge, which can be overlapped and delocalized with the CB minimum edge, leading to a reduced bandgap that may broaden the light absorption edge[19,20]. The XPS spectra show that the $Ag_{3d}$ peak of $Vs$-$SAL_{10}$ shifts to lower binding energy relative to that of SAL (Supplementary Fig. 10e), confirming the valance changes of Ag in $Vs$-$SAL_{10}$. The VB change of $AgInP_2S_6$ may lead to the corresponding changes of its CB[20]. The Mott–Schottky plots reveal that the CB edge of $V_S$-$SAL_{10}$ upshifts by ~0.06 and ~0.26 eV, relative to that of SAL and BC, respectively, as schematically illustrated in Supplementary Fig. 12. All BC, SAL, and $V_S$-$SAL_{10}$ were thus confirmed to possess suitable bandgaps as well as the appropriate band edge positions for photocatalytic $CO_2$ reduction under visible-light irradiation.

**Photocatalytic performance toward $CO_2$ photoreduction.** The photocatalytic $CO_2$ conversion was carried out in the presence of water vapor under simulated solar irradiation (Fig. 3). CO was detected the major product for BC and SAL (Fig. 3a, b). BC shows the CO yield of 2.44 $\mu mol\,g^{-1}$ for the first hour and a trace amount

of $CH_4$ of 0.63 $\mu mol\,g^{-1}$ (Fig. 3a). The photogenerated holes in the VB oxidize $H_2O$ to produce hydrogen ions by the reaction of $H_2O \rightarrow 1/2O_2 + 2H^+ + 2e^-$. CO is formed by reacting with two protons and two electrons $(CO_2 + 2e^- + 2H^+ \rightarrow CO + H_2O\,(1))$, and $CH_4$ formation through accepting eight electrons and eight protons $(CO_2 + 8e^- + 8H^+ \rightarrow CH_4 + 2H_2O\,(2))$. SAL exhibits 6.9 and 14.3-time enhancement of production of CO and $CH_4$ relative to BC, reaching 17.1 and 9.0 $\mu mol\,g^{-1}$ for the first hour, respectively (Fig. 3b). A small amount of $H_2$ was also generated as a typical competitive reaction with $CO_2$ reduction (Supplementary Fig. 13). The prerogative of atomic ultrathin geometry of SAL may be mainly responsible for the enhanced photocatalytic activity besides larger surface area, allowing charge carriers to move from interior to the surface quickly to conduct catalysis, avoiding the recombination in the body. A small amount of $C_2H_4$ was also detected for SAL with a yield of 5.3 $\mu mol\,g^{-1}$. $C_2H_4$ is generated by accepting 12 electrons and 12 protons $(2CO_2 + 12e^- + 12H^+ \rightarrow C_2H_4 + 4H_2O\,(3))$. With the $H_2O_2$ etching process, excitingly, $C_2H_4$ excitingly becomes the main product for $V_S$-$SAL_{10}$ with a yield of 44.3 $\mu mol\,g^{-1}$ (Fig. 3c). The calculated yield-based selectivity reaches ~73%, and the electron-based selectivity is as high as ~89%[21] (Fig. 3e). Meanwhile, CO and $CH_4$ minority products were also traced with the yields of 10.9 and 5.6 $\mu mol\,g^{-1}$, respectively, both less than the case of SAL. It indicates that the surface of $V_S$-$SAL_{10}$ preferentially promotes the $C_1$ intermediates to C–C couple into $C_2$ product rather than liberate them into free CO and $CH_4$ gases. The quantum yield of $V_S$-$SAL_{10}$ was measured 0.51% at a wavelength of 415 nm using monochromatic light (see the details in SI). The etching process time was found determinative for the dominant production of $C_2H_4$. The EPR measurement shows that the signal intensity gradually increases with prolonging etching time from 5 to 15 s (Supplementary Fig. 5), indicating being raised a number of $V_S$ in $V_S$-SAL. Elongation of the etching time from 5 to 10 s was favorable for increasing the yield of $C_2H_4$ (Supplementary Fig. 14). However, a much long etching time of 15 s decreases activity negatively, which may be due to that an excess of $V_S$ defects may accelerate the recombination of

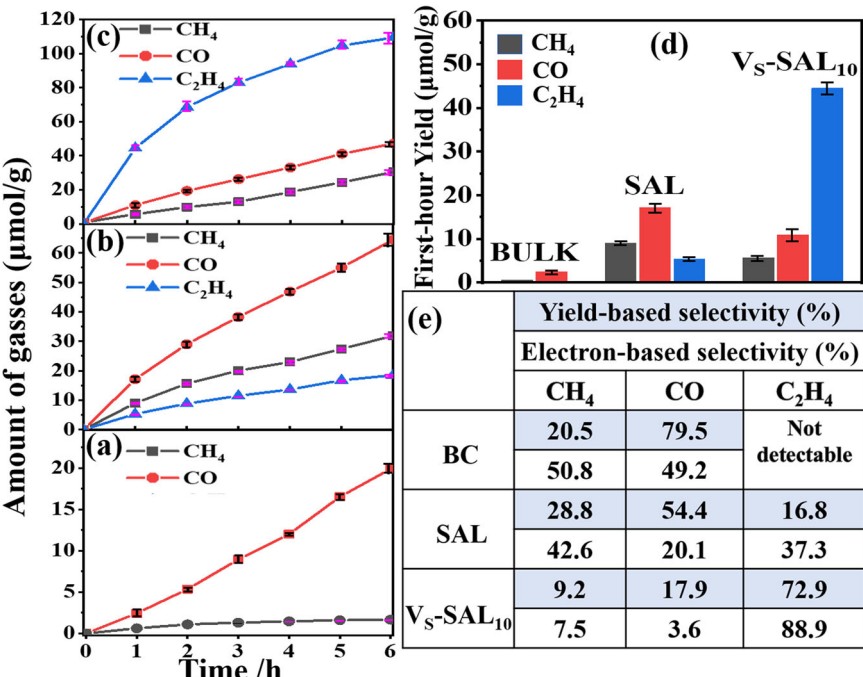

**Fig. 3 Photocatalytic $CO_2$ reduction performance.** Photocatalytic gases evolution amounts as a function of light irradiation times of **a** BC, **b** SAL, and **c** $V_S$-SAL. **d** Photocatalytic activity for the first hour. **e** Table illustration for the yield and electron-based selectivities of photocatalytic $CO_2$ conversion.

| (e) | Yield-based selectivity (%) | | |
|---|---|---|---|
| | Electron-based selectivity (%) | | |
| | $CH_4$ | CO | $C_2H_4$ |
| BC | 20.5 | 79.5 | Not detectable |
| | 50.8 | 49.2 | |
| SAL | 28.8 | 54.4 | 16.8 |
| | 42.6 | 20.1 | 37.3 |
| $V_S$-$SAL_{10}$ | 9.2 | 17.9 | 72.9 |
| | 7.5 | 3.6 | 88.9 |

photogenerated carriers[22]. The reduction experiment of $CO_2$ performed in the dark or absence of the photocatalyst shows no appearance of CO and hydrocarbon products, proving that the reduction reaction of $CO_2$ is driven by light under photocatalyst. A blank experiment with the identical condition and in the absence of $CO_2$ shows no appearance of $C_2H_4$, CO, and $CH_4$, proving that the carbon source was completely derived from input $CO_2$. An isotope labeling experiment using $^{13}CO_2$ confirms that the produced $C_2H_4$ originates from the input $CO_2$ (Supplementary Fig. 15). The $O_2$ production was also detected using the similar isotope $H_2^{18}O$ tracer control experiment (Supplementary Fig. 15). It should be mentioned that after 12 h light irradiation, increased tendency of the generation of the hydrocarbon products over the present photocatalyst slowed down. It may be assigned to the potential carbon deposition as intermediates covering the active sites of the photocatalyst during the photoreduction process. The problem may be resolved through post washing treatment to recover the catalytic activity to a certain extent, as shown in Fig. S16.

### Mechanism of the photocatalytic performance of the $V_S$-SAL.

DFT simulations were performed to explore the $V_S$-mediated catalytic selectivity mechanism toward CO and $C_2H_4$ on $AgInP_2S_6$. $CO_2$ molecules are initially adsorbed on the catalyst surface where $H_2O$ molecules dissociate into hydroxyl and hydrogen ions at the same time. The free-energy profile for the photocatalytic $CO_2$-to-hydrocarbon process with the lowest-energy pathway on the perfect $AgInP_2S_6$ surface was calculated, as shown in Fig. 4. The key step for CO production is the hydrogenation of $CO_2$ to form *COOH, and the free-energy change of the step is 0.48 eV. Subsequently, the reaction intermediate (*COOH) further couples a proton/electron pair to generate CO and $H_2O$ molecules. The adsorption energy of −0.07 eV of the produced *CO on the defect-free $AgInP_2S_6$ surface implies the physical adsorption on the catalyst (Supplementary Fig. 17a). It means that *CO molecules can easily

liberate from BC and SAL to become free CO gas, allowing high CO catalytic selectivity. Additional parts of *CO were continuously reduced by the incoming electrons and the successive protonation process to transform into $CH_4$[20,23]. While the charge density of the valence band (VB) for pristine $AgInP_2S_6$ is evenly located on all the S and Ag atoms, contrastingly, the charge density of the VB is mainly located on the Ag atoms near the $V_S$ for $V_S$-$AgInP_2S_6$, (Supplementary Fig. 18). That is to say, the presence of $V_S$ in $V_S$-$AgInP_2S_6$ causes the charge enrichment on the Ag atoms near the $V_S$, which would benefit for stabilizing the reaction intermediates. For $V_S$-SAL, $V_S$ can act as a trap for the *CO molecule, that is, the *CO molecule can chemically adsorb at exposed Ag sites with an adsorption energy of −0.25 eV (CO can only physically adsorb on the exposed P and In sites with a distance of 2.56 and 3.20 Å, See Supplementary Fig. 17b–d). The higher CO onset desorption temperature on $V_S$-$SAL_{10}$ than SAL affirms the stronger absorption (Supplementary Fig. 19). The absorbed *CO can be further protonated to successively form a series of key reaction intermediates with unsaturated coordination, which was confirmed with in situ FTIR measurement (Supplementary Fig. 20). The other *CO molecules produced on the surface diffuses toward $V_S$ and couple with those reaction intermediates to produce $C_2H_4$. The $C_2H_4$ free energy diagrams are summarized in Fig. 4c, while the corresponding C–C coupling barriers are presented in Fig. 4b. The different C-C coupling energy barriers were evaluated for three unsaturated reaction intermediates (*COH, *CHOH, and *CH$_2$) (Fig. 4b). The coupling energy barrier with a value of 0.84 eV (*CO–CHOH) is lower than that of other coupling pathways (*CO–COH, 1.01 eV and *CO–CH$_2$, 1.84 eV), hence the $C_2H_4$ will be produced via CO–CHOH coupling and hydrogenation. The whole free energy diagram shows that the process of *CO to *COH is regarded as the potential determining step (0.86 eV). It should be especially emphasized that the detected small amount of $C_2H_4$ on SAL possibly originates from the potential existence of the tiny number of $V_S$ in SAL, resulting from mechanically detaching

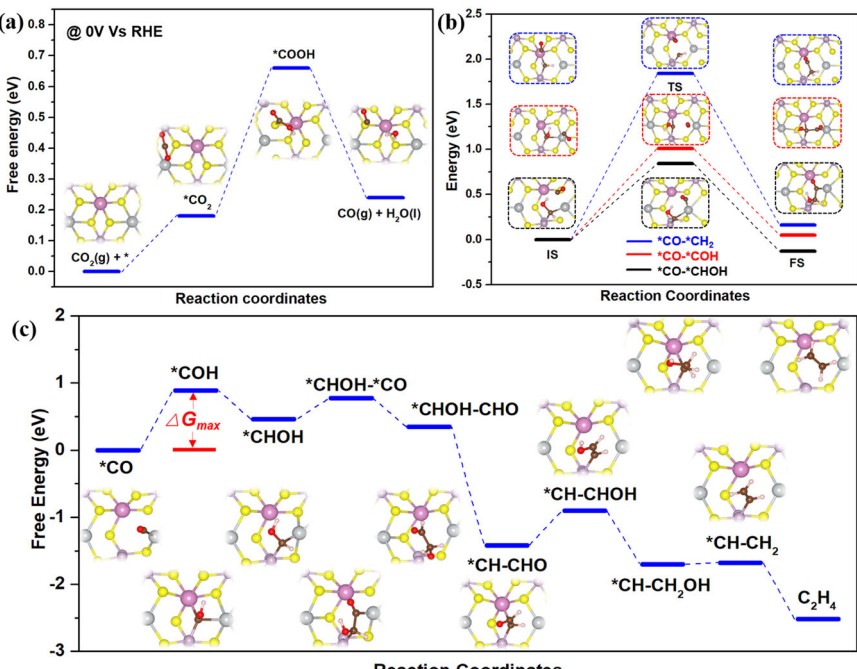

**Fig. 4 Theoretical investigations. a** Gibbs free energy diagrams for $CO_2$ reduction to CO over perfect $AgInP_2S_6$. **b** Three kinds of possible C–C coupling pathways over $AgInP_2S_6$ containing $V_s$. **c** Gibbs free energy diagrams for CO reduction to $C_2H_4$ over $AgInP_2S_6$ with $V_s$. The insets show the corresponding optimized geometries for the reaction intermediates during the $CO_2$ reduction process. Sulfur, phosphorus, indium, silver, carbon, oxygen, and hydrogen atoms are yellow, purple, lilac, gray, black, red, and white, respectively.

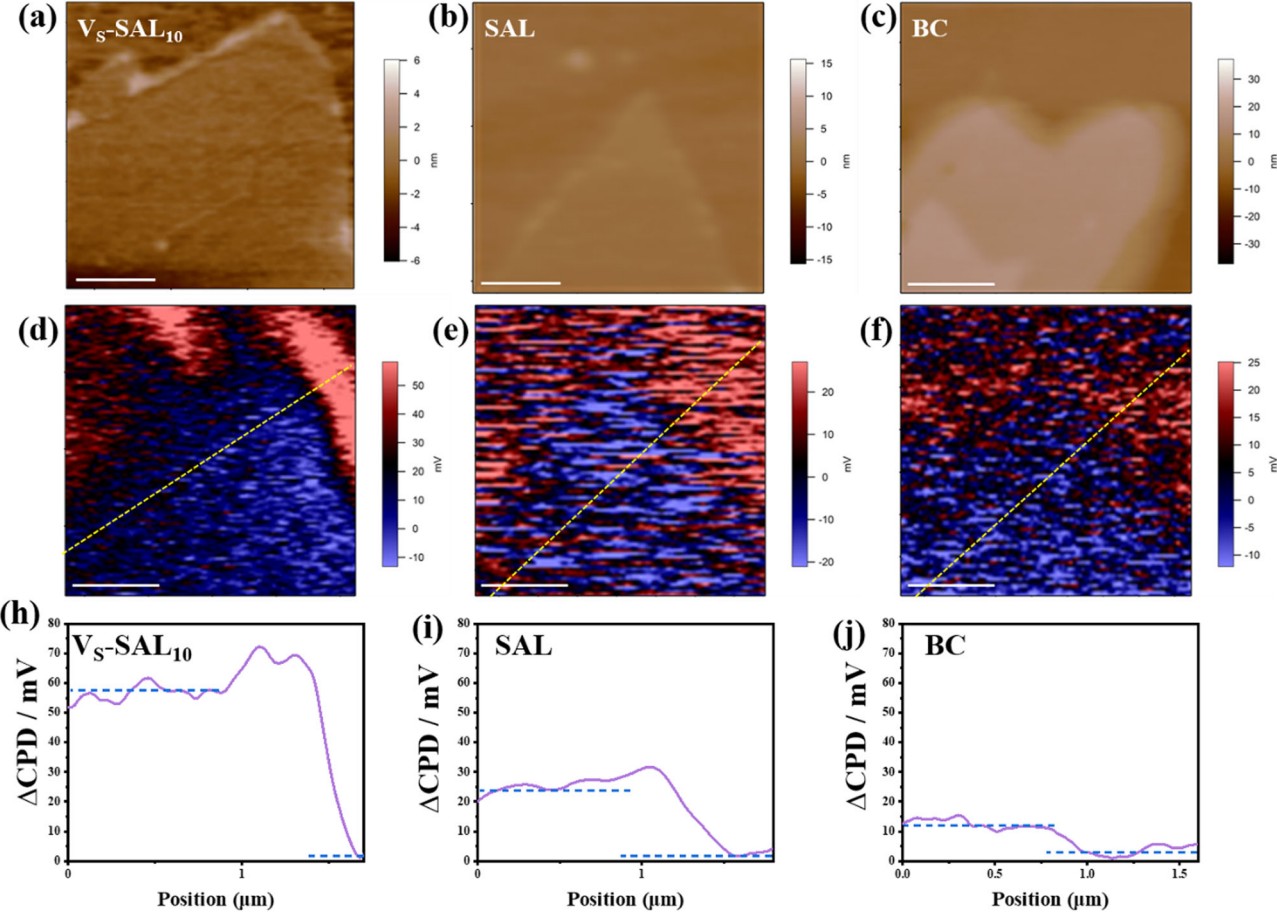

**Fig. 5 SPV characterization.** Height images of **a** $V_S$-$SAL_{10}$, **b** SAL, and **c** BC. The SPV images **d** $V_S$-$SAL_{10}$, **e** SAL, and **f** BC in (**a**–**c**), respectively, are differential images between potential images under light and in the dark. All scale bars represent 0.5 μm. The surface photovoltage change by subtracting the potential under dark conditions from that under illumination (SPV, $\Delta CPD = CPD$ dark $− CPD$ light) of **h** $V_S$-$SAL_{10}$, **i** SAL, and **j** BC.

sulfur atoms from SAL during the probe sonication exfoliation process. The reaction process for the reduction of $CO_2$ into $C_2H_4$, CO, and $CH_4$ over $V_S$-SAL under light illumination is thus proposed in Supplementary Fig. 21. To confirm CO as an important intermediate for the $C_2H_4$ formation, CO as the starting reactant substituting for $CO_2$ was also conducted for the similar photocatalytic performance. The result reveals that a considerable amount of $C_2H_4$ was indeed detected (Supplementary Fig. 22). In addition, a small amount of ethane ($C_2H_6$) and propylene ($C_3H_6$) were also produced. It indicates that CO as starting reactants may be further favorable for C–C, even $C_2$–C coupling.

Surface photovoltage spectroscopy (SPV) was employed to study the separation and transport behavior of photoinduced charge carriers of the studied $AgInP_2S_6$. More negative SPV signal change reflects a higher concentration of photogenerated electrons before and after light illumination. All BC, SAL, and $Vs$-$SAL_{10}$ show the SPV response under light illumination (Fig. 5 and Supplementary Fig. 23), corresponding to band-to-band transition. The SAL and BC exhibit 20–30 mV and 5–10 mV negative change before and after light illumination, respectively. More negative SPV signal change of SAL than BC exactly demonstrates that the atomically thin structure enables to alleviate the bulk electron–hole recombination to achieve high-concentration accumulation of photogenerated electrons on the surface. The $V_S$-$SAL_{10}$ display obviously dramatic change of 50–60 mV, indicating that introduction of $V_S$ can further favor the carrier separation and allow much increment of electron

concentration on the surface. The excess surviving electrons are not only the necessary prerequisite to photoconversion of $CO_2$, but also can promote $CO_2$ adsorption and activation on the surface of the photocatalyst.

Photoluminescence (PL) decay profiles show that the SAL (~1.32 ns) possesses a longer PL lifetime than BC (~0.40 ns) (Supplementary Fig. 24), demonstrating that the atomically thin structure can indeed shorten the transfer distance of the carriers and decrease recombination chance of electron and hole in the body. $V_S$-$SAL_{10}$ exhibits the longest PL lifetime (~1.50 ns), confirming that the surface $V_S$ can serve as surface separation centers for charge carriers and further promote the charge separation, therefore offering more opportunities for photocatalytic $CO_2$ reduction. Transient photocurrent shows that the photocurrent intensity of SAL was enhanced with a steadily repeating course due to promoted charge separation, compared with BC (Supplementary Fig. 25a). The highest photocurrent intensity of $V_S$-$SAL_{10}$ implies that the $V_S$ also makes an effective contribution to saving carriers. Electrochemical impedance spectra reveal that $V_S$-$SAL_{10}$ manifests the smallest semicircle in Nyquist plots (Supplementary Fig. 25b), suggesting the lowest charge-transfer resistance, which permits fast transport of photoinduced charge.

## Discussion
In summary, single atomically thin $AgInP_2S_6$ layers were successfully synthesized through a facile probe sonication exfoliation

of BC. The atomically thin structure of SAL, relative to BC, enables more charge carriers to mobile from the interior onto the surface and surviving accumulate onto the active sites to improve the photocatalytic activity. While SAL exhibits obvious conversion efficiency with CO as the major product, the presence of $V_S$ in $V_S$-SAL changes the $CO_2$ photoreduction pathway to allow the dominant generation of $C_2H_4$. This work not only paves an effective approach for selectively producing multi-carbon products from $CO_2$ photoreduction but also provides a new insight for catalyst design through vacancy defect engineering.

## Methods

**Synthesis of BC, SAL, and $V_S$-SAL**. The $AgInP_2S_6$ crystals have been synthesized by physical vapor transport (PVT) in a two-zone furnace. Stoichiometric amounts of high-purity elements (mole ratio Ag: In: P: S = 1:1:2:6, around 1 g in total) were sealed into a quartz ampoule with the pressure of $1 \times 10^{-4}$ Torr inside the ampoule. The length of the quartz ampoule was about 15–18 cm with a 13 mm external diameter. The ampoule was kept in a two-zone furnace ($680 \rightarrow 600\,°C$) for 1 week[11]. After the furnace was cooled down to room temperature, the $AgInP_2S_6$ crystalline powders could be found inside the ampoule (Supplementary Fig. 1a). SAL was prepared by sonication-assisted liquid exfoliation processes from synthetic $AgInP_2S_6$ crystalline powders. For the detail, $AgInP_2S_6$ bulk powder was ground carefully, and then dispersed in ethanol solution. After continuous ultrasonification for 12 h with a probe-type cell crusher (~1200 W), the solution was conducted for static settlement, and the supernatant was taken to ultrasonic dissection further. Through $4000 \times g$ for 10 min centrifugation, the samples peeled insufficiently were removed, and the supernatant was collected through an additional $12,000 \times g$ for 20 min centrifugation to obtain the $AgInP_2S_6$ monolayer. The derived SAL has dispersed in the water again for subsequent liquid nitrogen refrigeration and being dried in a vacuum freezing dryer at a pressure below 20 Pa for 2 days. The residual ethanol can be considered to be totally removed.

SAL was immersed in $H_2O_2$ solutions with the of concentrations 0.1 mol/L inside which SAL was allowed to react with $H_2O_2$ for 5, 10, and 15 s, referred to $V_S$-SAL$_5$, $V_S$-SAL$_{10}$, and $V_S$-SAL$_{15}$, respectively, at 25 °C. All the obtained samples were carefully washed and dried before use.

**Characterizations**. XRD (Rigaku Ultima III, Japan) was used to investigate the purity information and crystallographic phase of the as-prepared powder samples. The XRD pattern was recorded by using Cu-ka radiation ($\lambda = 0.154178$ nm) at 40 kV and 40 mA with a scan rate of $10°\,min^{-1}$. The morphology was characterized by the FESEM (FEI NOVA NANOSEM 230). The TEM and HRTEM images were taken on a JEM 200CX TEM apparatus. X-ray photoelectron spectroscopy (XPS; K-Alpha, Thermo Fisher Scientific) was standardized according to the binding energy of the adventitious C 1s peak at 284.8 eV, which was used to inspect the chemical states. A UV–vis spectrophotometer (UV-2550, Shimadzu) was hired to record the UV-visible diffuse reflectance spectra and switched to the absorption spectrum on the basis of the Kubelka−Munk connection at room temperature. In situ FTIR spectra were measured with synchronous illumination Fourier transform infrared spectroscopy on Bruker IFS 66V FT spectrometer. The PL decay profile was described by the single-particle confocal fluorescence spectroscopy measurement (PicoHarp300). SPV was detected through AFM (Asylum Research, MFP-3D-SA, USA) analysis with the photo-assisted (a 405 nm laser excitation) Kelvin probe force microscopy. Photoelectrochemical measurements were detected by a CHI660E electrochemical workstation using a standard three-electrode system in 1 mM $NaSO_4$ solution. Soft X-ray absorption spectra (XAS) were collected from the Soft X-ray Spectroscopy beamline at the Australian Synchrotron (AS, Australia), part of ANSTO.

For the electrochemistry measurement, the $AgInP_2S_6$ catalyst ink was prepared by dispersing 10 mg of as-prepared catalysts in 1 mL of ethanol under sonication. Then, 50 μL of the ink was evenly spread onto a piece of pretreated FTO within a 1 $cm^2$ area and dried at room temperature. The catalysts were thus attached to FTO. The solid-state current–voltage ($J–V$) test curves exhibit Ohmic characteristics (Supplementary Fig. 26), confirming the formation of ohmic back contact between samples and FTO. The working area of the electrode is as large as 1 $cm^2$. The scan rate was 5 mV $s^{-1}$. The reference electrode was the saturated Ag/AgCl electrode, and a Pt foil was employed as the counter electrode. The 0.5 M $Na_2SO_4$ aqueous solution was used as the electrolyte.

**Measurement of photocatalytic activity**. For the photocatalytic reduction of $CO_2$, 4–5 mg of sample was uniformly dispersed on the glass reactor with an area of 4.2 $cm^2$. A 300 W Xenon arc lamp was used as the light source of the photocatalytic reaction. The volume of the reaction system was about 460 ml. Before the irradiation, the system was vacuum-treated several times, and then the high purity of $CO_2$ gas was followed into the reaction setup for reaching ambient pressure. Totally, 0.4 mL of deionized water was injected into the reaction system as a reducer. The as-prepared photocatalysts were allowed to equilibrate in the $CO_2/H_2O$ atmosphere for several hours to ensure that the adsorption of gas molecules was complete. During the irradiation, about 1 mL of gas was continually taken from the reaction cell at given time intervals

for subsequent CO, $CH_4$, and $C_2H_4$ concentration analysis by using a gas chromatograph (GC-2014C, Shimadzu Corp., Japan).

**The external quantum efficiency (EQE)**. The quantum yield was calculated according to the below equation

$$E_Q = N(\text{electron})/N(\text{photon})$$
$$= [N(CO) \times 2 + N(CH_4) \times 8 + N(C_2H_4) \times 12]/N(\text{photon}) \times 100\% \quad (4)$$

where $N$ (electron) signifies two electrons are required to produce one molecule CO in unit time. The $N$ (photon) is figured out according to the equation:

$$N(\text{photon}) = [\text{light intensity} \times \text{illumination area} \times \text{time}]/[\text{average single photon energy} \times N_A] \quad (5)$$

Light-emitting diodes (LEDs) provides the monochromatic incident light with identical conditions. The light intensity of LEDs with 415 nm wavelength is 10.5 mW/$cm^2$, the illumination area is controlled to 4.91 $cm^2$, $N_A$ is the Avogadro constant, and the average single photon energy is calculated according to the equation:

$$E(\text{photon}) = hc/\lambda \quad (6)$$

in which $h$ is the Planck constant, c indicates the speed of light, and $\lambda$ is the wavelength.

**Computational details**. The density functional theory (DFT) calculations were made with the Vienna Ab Initio Simulation Package[24,25] code. The exchange-correlation interactions and the ion–electron interactions were solved by the Perdew–Burke–Ernzerhof functionals[26,27] and the projector-augmented wave method[28], respectively. The monolayer $AgInP_2S_6$ was a model with a $2 \times 2$ super-cell. A plane-wave cutoff of 450 eV was adopted and the maximal force on all-atom was below 0.02 eV/Å. The distance between periodic units in the vertical direction was larger than 16 Å. The DFT-D2 method of Grimme[29] was used in all calculations to accurately describe long-range Van der Waals (vdW) interactions. The climbing-image nudged elastic band (CI-NEB) method[30] incorporated with spin-polarized DFT was used to locate the minimum-energy path. The intermediate images of each CI-NEB simulation were relaxed until the perpendicular forces were smaller than 0.1 eV/Å.

The free energies of each reaction intermediates were determined according to $G = E + ZPE - TS$. (7) The electronic energies ($E$) can be directly obtained from DFT computations. The zero-point energy (ZPE) and entropy correction (TS) were calculated from vibration analysis by standard methods. The computational hydrogen electrode model[31] was used to treat the free energy change of each reaction step involving a proton–electron pair transfer. In this model, the free energy of a proton-electron pair at 0 V vs. RHE is equal to half of the free energy of a hydrogen molecule.

## Data availability

The data that support the findings of this study are available from the corresponding author upon reasonable request.

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

## Acknowledgements

The authors wish to acknowledge the support of the National Key R&D Program of China (2018YFE0208500), 973 programs (2017YFA0204800), NSF of China (21972065, 21773114, and 21773027), the Fundamental Research Funds for the Central University (020414380167), NSF of Jiangsu Province (No. BK20171246), the Hefei National Laboratory for Physical Sciences at the Microscale (KF2020006), the Program for Guang-dong Introducing Innovative and Entrepreneurial Team (2019ZL08L101) and The University Development Fund (UDF01001159). Prof. Ran Long and Dr. Wenqing Zhang of USTC (China) were greatly acknowledged for in situ FT-IR measurements.

## Author contributions

Y.Z., Y.X. and Z.Z instructed this work. L.S. and J.W. carried out the DFT calculation. W.G., H.H., X. Li, Z.C., Y.Y. and Q.S. performed the experiments and co-wrote this paper. X.W. contributed to the PL spectrum measurement.

## Competing interests

The authors declare no competing interests.
