## [Peer Review File · Nature Communications]

REVIEWER COMMENTS

Reviewer #1 (Remarks to the Author):

The authors describe in their publication entitled “Vacancy-Defect Modulated Pathway of Photoreduction of CO₂ on Quaternary Single Atomically Thin AgInP₂S₆ Sheets toward Boosting Efficient and Selective Production of Value-Added Olefiant Gas” the synthesis and testing of defect quaternary AgInP₂S₆ in the photoreduction of CO₂. The manuscript itself is drafted in a logical way but cannot be considered for publication in Nature Communication due to the following most important issues: The authors present results from a highly important research field, but their lack of interpretation of the obtained results makes a publication in this journal not feasible. The authors state they exfoliated atomic layer sheets of ca. 7 Å, but there is no unambiguous proof of this assumption. Neither the synthesis protocol nor the interpretation of the obtained analytic data (eg. Figure 1a) support this assumption. How is it possible to mechanically exfoliate by ultrasound such thin sheets from polycrystalline material? The interpretation of the TEM images is highly doubtful to the arbitrary selection of parts from the nanostructure (for example Figures 1c and 1e). In general, the imaging of the nanomaterial is insufficient due to incomparable resolutions and erroneous interpretation thereof. The results from photoreduction themselves are doubtful, too. Especially the material supposedly designated as ultrathin was extensively treated with ethanol, which itself as trace impurity in the material could lead to higher CH₄, CO and C₂H₄ values (see Figure 3). There are completely no investigations under conditions of highest purity to proof the origin of the carbon in the resulting products. The authors should provide comparable measurements under highest achievable purity to ensure product formation from CO₂. Therefore, at the current stage of research I must propose rejection of the manuscript.

Reviewer #2 (Remarks to the Author):

The photocatalytic conversion of CO₂ into C₁ species has been reported largely in many studies. In this paper, the authors show that the hydrogenation of CO₂ into ethylene on AgInP₂S₆ ultrathin sheet photocatalysts. This is an interesting result. However, the evidence for the catalytic processes of ethylene production is insufficient, and some characterization results is confusing. Thus, I think the paper should be made a major revision to clarify the following question before consideration for publication.

(1) EPR and HAADF-STEM is seems to well verify the formation of Vs defects on the H₂O₂ treated sample, but it should be noted that the EPR signal for the possible existence of defect sites in samples can be come from the open of S-M(metal) bond or the insertion of O atom. On the other hand, due to the single atomically-thin samples, the metal atom is easily exposed on the outermost surface, the HAADF-STEM images of the parent SAL should be provided for the comparison. Moreover, the atomic resolution HAADF-STEM image will be better to confirm the defect structure.

(2) Besides the introduction of defects, are there any other changes in the structure and composition of sample with H₂O₂ treatment? For example, crystal structure or the oxidation of samples. Thus, XRD pattern and O 1S XPS spectra of VS-SAL10 samples should be necessary.

(3) on the one hand, the author deduced that introduction of VS induces the impurity states near the conduction band (CB) edge to narrow the bandgap of SAL. On the other hand, the schematic electronic band structures from the Mott-Schottky plots show a higher conduction band edge of VS-SAL10 than that of SAL. Obviously, the role of VS on the electronic structure is not clear in this paper.

(4) Error bars should be reported for the data in Figure 3.

(5) Hydrogen production from water reduction is a typical competitive reaction with carbon dioxide reduction, which is completely ignored in this work.

(6) It is well known that the sulfide photocatalyst often have fatal stability problems. How about the photocatalytic cycle reaction of CO₂ conversion and the chemical stability of VS-SAL10 samples for long time reaction.

(7) In Figure S14, desorption products should be detected with Mass spectrum to judge whether desorption products are related to CO molecular.

(8) In Figure S15, the baseline fluctuation of infrared spectrum is very serious, the assignment of weak IR peak is problematic. It is also confused that the IR peaks is almost disappeared with 30 min light irradiation on BC and SAL samples. The IR peaks of C₂H₄ products is also observed on BC, but BC samples is not observed to yield C₂H₄ in the photocatalytic performance. Moreover, No IR peaks for H₂O reactants at ~1600 cm⁻¹ is observed, why?

(9) In the proposed research pathway for C₂H₄ formation, CO was considered as an important intermediate for the further C₂H₄ formation by the DFT calculation. Can it be confirmed by experiment, such as using CO as the starting reactants for the photocatalytic performance?

Reviewer #3 (Remarks to the Author):

This is an interesting and potentially quite important manuscript. The concept that one can design surface sites that direct the coupling of carbons in CO₂ to form C₂⁺ species is an important advance in CO₂ chemistry and heterogeneous catalysis in general. Overall the science and conclusions presented are of good quality. However, prior to publication, in my opinion, there are certain aspects of the paper that need to be revised/edited:

- This paper needs to be edited by a native speaker. Beyond simple issues of grammar and word choice,

there are several sentences in the manuscript that I cannot understand and I'm certain don't mean to convey the meaning that the selected words suggest.

- On the one hand, the authors indicate that their material is essentially one unit cell thick. But, they also repeatedly talk about surface vs. bulk states. How can one have bulk states if there is no bulk?
- Neither in the manuscript or the SI are sufficient details given to reproduce the electrochemistry that is reported.

o How is the electrode made? What material is the AgInP₂S₆ attaching to?

o How did the researchers assure (or test for) the formation of an ohmic back contact? What was the working area of the electrode?

o What scan rate(s) was used in Figure S9? What reference electrode and counterelectrode were employed? What is the composition of the electrolyte?

o Figures S9a-c are referred to as Mott-Shottky plots. However, that should be a plot of $1/C^2$ vs. potential. The units on the vertical axis are not correct for MS plots. It would also help if the intercepting potentials shown on the figures were labeled "Flatband Potential".

o Figure S11 is called "Gas chromatogram and mass spectra...". But, there is only ms data. No gc data is provided. Figure S12 is referred to in the S11 figure caption. But, the discussion does not go with the S12 figure that is provided. It looks like a figure might be missing.

Eco-materials and Renewable Energy Research Center (ERERC)
National Laboratory of Solid State Microstructures
Nanjing University

Tangzhong Ying Building C412, Nanjing University, 22 Hankou Road, Nanjing, Jiangsu 210093,
P. R.China: +86-25-8362-1372 E-mail: zhoyong1999@nju.edu.cn

Response to the reviewers' Comments

Reviewer #1:

The authors describe in their publication entitled “Vacancy-Defect Modulated Pathway of Photoreduction of CO₂ on Quaternary Single Atomically Thin AgInP₂S₆ Sheets toward Boosting EfficientThe manuscript itself is drafted in a logical way but cannot be considered for publication in Nature Communication due to the following most important issues.

We are much regretful for this reviewer's rejection decision for our manuscript. In our work, the quaternary AgInP₂S₆ atomic layer with the thickness of ~ 0.70 nm were successfully synthesized through facile ultrasonic exfoliation of the corresponding bulk crystal. The ultrathin sheet exhibits efficiently photocatalytic conversion of CO₂ into CO as a major product and minority of CH₄ and C₂H₄ in the presence of water vapor. The sulfur defect engineering on this atomic layer through a H₂O₂ etch process can excitingly enable to change the CO₂ photoreduction reaction pathway to steer dominant generation of C₂H₄ important chemical with the yield-based selectivity reaching ~73% and the electron-based selectivity as high as ~89%, and the quantum yield of 0.51% at wavelength of 415 nm. We believe that this work is significant and may provide fresh insights into the design of atomically thin photocatalyst framework for CO₂ reduction and establish an ideal platform for reaffirming the versatility of defect engineering in tuning catalytic activity and selectivity, which will receive broad interest for the readers of *Nat. Commun.*.

1. As to *“The authors present results from a highly important research field, but their lack of interpretation of the obtained results makes a publication in this journal not feasible. The authors state they exfoliated atomic layer sheets of ca. 7 Å, but there is no unambiguous proof of this assumption. Neither the synthesis protocol nor the interpretation of the obtained analytic data (eg. Figure 1a) support this assumption. How is it possible to mechanically exfoliate by ultrasound such thin sheets from polycrystalline material? The interpretation of the TEM images is highly doubtful to the arbitrary selection of parts from the nanostructure (for example Figures 1c and 1e). In general, the imaging of the nanomaterial is insufficient due to incomparable resolutions and erroneous interpretation thereof.”*

It is well known that ultrasound can transfer high energy into layered materials and weaken the Van der Waals forces between adjacent layers, which results in effective delamination. Actually, considerable number of single or few layer materials have been successfully prepared in recent years through the ultrasonic liquid-phase exfoliation techniques, such as ultrathin Mg–Al LDHs structures with thickness at around 0.8 nm (*Adv. Eng. Mater.* **2017**, *7*, 1601657), monolayer BiO_{2-x} (*Angew. Chem. Int. Ed.* **2018**, *57*, 491; *Nat. Commun.* **2019**, *10*, 788), single-layered MoS₂ (*Nat. Chem.* **2017**, *9*, 810), BiOBr atomic layers with the approximately thickness of 0.81 nm (*Angew. Chem. Int. Ed.* **2018**, *57*, 8719). Anyway, more than 1,200 related papers about ultrasonification-based exfoliation for the preparation of monolcan be found in Web of Science with the keywords of **ultrasonic*** and **monolay***, as presented blow.

Web of Science

Search

Results: 1,211

(from All Databases)

You searched for: TOPIC:
(ultrasonic* and monolay*)

To further confirm the monolayer formation of the present AgInP₂S₆ nanosheet convincingly, the additional more AFM images of SAL are provided in the revision (Please see Figure S4 in the revision), accurately demonstrating 0.6-0.7 nm thickness of the resulting AgInP₂S₆ ultrathin sheet. Furthermore, the more detail preparation method is also added in the revision (See experimental section), described as following: “In the preparation process of SAL, AgInP₂S₆ bulk powder was ground carefully, and then dispersed in ethanol solution. After continuous ultrasonification for 12 h with a probe-type cell crusher (1,200 W), the solution was conducted for static settlement, and the supernatant was taken to ultrasonic dissection further. Through 4,000 rpm for 10 min centrifugation, the samples peeled insufficiently were removed, and the supernatant was collected through additional 12000 rpm for 20 min centrifugation to obtain the AgInP₂S₆ monolayer.”

Figure S4. AFM images of SAL.

2. As to *“The results from photoreduction themselves are doubtful, too. Especially the material supposedly designated as ultrathin was extensively treated with ethanol, which itself as trace impurity in the material could lead to higher CH₄, CO and C₂H₄ values (see Figure 3). There are completely no investigations under conditions of highest purity to proof the origin of the carbon in the resulting products. The authors should provide comparable measurements under highest achievable purity to ensure product formation from CO₂. Therefore, at the current stage of research I must propose rejection of the manuscript.”*

We are still much regretful for the decision. After the exfoliation steps of the samples (ultrasonic liquid-phase exfoliation, static settlement, and centrifugation), the derived SAL was dispersed in water again for subsequent liquid nitrogen refrigeration and being dried in a vacuum freezing dryer at pressure below 20 Pa for two days. The

residual ethanol can be considered to be totally removed, which was further proved with the FTIR spectra, showing no typical peaks at around 1050 cm^{-1} for ethanol.

Figure FTIR spectra of SAL and V_S-SAL, showing no peak assigned to ethanol residual (Only for review).

In addition, a blank experiment with identical photocatalysis condition and in the absence of CO₂ shows no appearance of C₂H₄, CO, and CH₄, proving that the carbon source was completely derived from input CO₂. An isotope labeling experiment using ¹³CO₂ definitely confirms that the produced C₂H₄ originates from the input CO₂.

Reviewer #2:

The photocatalytic conversion of CO₂ into C1 species has been reported largely in many studies. In this paper, the authors show that the hydrogenation of CO₂ into ethylene on AgInP2S6 ultrathin sheet photocatalysts. This is an interesting result. However, the evidence for the catalytic processes of ethylene production is insufficient, and some characterization results is confusing. Thus, I think the paper should be made a major revision to clarify the following question before consideration for publication.

1. As to “EPR and HAADF-STEM seems to well verify the formation of Vs defects on the H₂O₂ treated sample, but it should be noted that the EPR signal for the possible existence of defect sites in samples can be come from the open of S-M(metal) bond

or the insertion of O atom. On the other hand, due to the single atomically-thin samples, the metal atom is easily exposed on the outermost surface, the HAADF-STEM images of the parent SAL should be provided for the comparison. Moreover, the atomic resolution HAADF-STEM image will be better to confirm the defect structure.”

The Raman spectra show that the relative peak intensity of S-P-P and P-S-P to that of P-P for V_S -SAL were both lowered, compared with SAL (Figure S6). It may additionally verify that the detected defect sites can be assigned to V_S (*Phys. Rev. B* **2013**, 88, 035426), rather than the open of S-M(metal) bond or the possible insertion of O atoms.

Figure S6. Raman spectra of SAL and V_S -SAL.

The HAADF-STEM image of the parent SAL was provided for the comparison. Few bright spots assigned to V_S can be also observed (Figure S9a), which originates from that little amount of the metal atoms can be easily exposed on the outermost surface during the ultrasonic exfoliation process. With regard to the atomic resolution HAADF-STEM image, we have tried to our best effort for measurement, and still failed

to obtain the high-quality image. The Figure S9b is the best image that we can provide with our current technology. Two reasons may be responsible for the failure of obtaining ideal images. One is relatively no enough high resolution of the spherical aberration-corrected scanning TEM employed. The other is the intrinsic thermal instability of $V_s\text{-SAL}_{10}$. At high-magnification focusing operation of the STEM, the AgInP_2S_6 monolayer was observed to be easily molten and destroyed under electron-beam irradiation. We sincerely hope that the reviewer can understand our hard effort.

Figure S9. (a) HAADF-STEM images of the parent SAL. (b) High solution HAADF-STEM images of the parent SAL (Only for review).

2. As to “*Besides the introduction of defects, are there any other changes in the structure and composition of sample with H_2O_2 treatment? For example, crystal*

structure or the oxidation of samples. Thus, XRD pattern and O 1S XPS spectra of V_S-SAL₁₀ samples should be necessary.”

No obvious difference of the XRD patterns between SAL and V_S-SAL demonstrates no crystal structure change of the SAL before and after H₂O₂ etching treatment (Figure S2). Weak O1s XPS peaks were observed for both SAL and V_S-SAL₁₀, which more likely originate from absorbed components from ambience, such O₂ (Figure S10d). The almost same intensity and location of O1s peak confirms no apparent oxidation change before and after H₂O₂ treatment.

Figure S2. XRD patterns of BC, SAL and V_S-SAL₁₀.

Figure S10d. High-resolution O 1s XPS spectra of SAL and V_S -SAL₁₀.

3. As to “*on the one hand, the author deduced that introduction of V_S induces the impurity states near the conduction band (CB) edge to narrow the bandgap of SAL. On the other hand, the schematic electronic band structures from the Mott-Schottky plots show a higher conduction band edge of V_S -SAL₁₀ than that of SAL. Obviously, the role of V_S on the electronic structure is not clear in this paper.*”

We much appreciate this good question. Several literatures have also reported the similar change of the band structure, such as “ V_S -rich $CuIn_5S_8$ atomically thin layer (*Nature Energy* **2020**, 4, 690)” and “ V_O -rich WO_3 atomic layer (*Joule* **2018**, 2, 1004.),” in which introduction of sulfur or oxygen vacancies also induces the impurity states near the CB edge to narrow the bandgap, and simultaneously with a higher CB edge, as schematically illustrated below.

Band structures of the pristine CuIn_5S_8 and $\text{V}_\text{S}\text{-CuIn}_5\text{S}_8$ single-unit-cell layers and single unit-cell layers. (Reprinted from *Nature Energy* **2020**, 4, 690)

Band Structure for the V_O -Poor WO_3 atomic layers and WO_3 atomic layers. (Reprinted from *Joule* **2018**, 2, 1004.)

In the presence case, the DFT simulation suggested that the VB of pristine AgInP_2S_6 is comprised of the orbits of S and Ag atoms, while the VB of $\text{V}_\text{S}\text{-AgInP}_2\text{S}_6$ is mainly located on the Ag atoms near the V_S . The XPS investigation revealed that the Ag_{3d} peak of $\text{V}_\text{S}\text{-SAL}_{10}$ shifts to lower binding energy relative to that of SAL (Figure S10e), confirming the valance changes of Ag in $\text{V}_\text{S}\text{-SAL}_{10}$. Therefore, the VB change of AgInP_2S_6 may lead to the corresponding changes of its CB. Frankly, we do not fully understand the role of V_S on the electronic structure at presence. However, we believe that it is an interesting topic which deserves to be deeply investigated for us through

various transient spectrometer and steady-state fluorescent spectrograph. The detail result will be reported in our later work. We sincerely hope that the reviewer can understand our situation.

Figure S10e High-resolution Ag 3d XPS spectra of BC, SAL, and V_S-SAL₁₀.

4. As to *“Error bars should be reported for the data in Figure 3.”*

The error bars have been added in the revision.

5. As to *“Hydrogen production from water reduction is a typical competitive reaction with carbon dioxide reduction, which is completely ignored in this work.”*

Yes, the small amount of hydrogen gas from water reduction was also detected in the revision (Figure S13). SAL exhibited higher H₂ yield than V_S-SAL, which may be due to more preferring to absorption and activation of H₂O molecules on SAL. BC displayed the lowest H₂ production yield, which may be caused by few active sites and fast carrier recombination.

Figure S13. Photocatalytic H₂ evolution amounts as a function of light irradiation times over BC, SAL, and V_S-SAL₁₀.

6. As to *“It is well known that the sulfide photocatalyst often have fatal stability problems. How about the photocatalytic cycle reaction of CO₂ conversion and the chemical stability of V_S-SAL₁₀ samples for long time reaction.”*

Frankly, the stability of the present catalyst is not enough satisfying after long time irradiation. After 12 h light irradiation, increase tendency of the generation of the hydrocarbon products slowed down. It may be assigned to the potential carbon deposition as intermediates covering the active sites of the photocatalyst during photoreduction process. The problem may be resolved through post washing treatment to recover the catalytic activity, which is under investigation for later practical application.

7. As to *“In Figure S14, desorption products should be detected with Mass spectrum to judge whether desorption products are related to CO molecular.”*

Yes, we detected the desorption products with TPD-MS. The result proves that the desorption product is related to CO molecules.

8. As to “*In Figure S15, the baseline fluctuation of infrared spectrum is very serious, the assignment of weak IR peak is problematic. It is also confused that the IR peaks is almost disappeared with 30 min light irradiation on BC and SAL samples. The IR peaks of C₂H₄ products is also observed on BC, but BC samples is not observed to yield C₂H₄ in the photocatalytic performance. Moreover, No IR peaks for H₂O reactants at ~1600 cm⁻¹ is observed, why?*”

Yes, we agree with the serious signal-noise ratio of the present *in-situ* FTIR spectra, which is due to the low resolution of the employed FTIR instrument. In this revision, synchronous illumination diffuse reflectance Fourier transform infrared spectroscopy (SIDRIFTS) on Bruker IFS 66V FT spectrometer was employed to pursue to obtain the high-quality *in-situ* FTIR spectra. It is clear that the signal-noise ratio of the spectra has been improved much. The characteristic peaks corresponding to the various crucial intermediates for reducing CO₂ to C₁ and multi-carbon products can be detected clearly (Figure S19). Those peak intensities also gradually increase with extension of the irradiation time for BC, SAL, and V_S-SA₁₀.

With regard to appearance of the weak peak assigned to C₂H₄ on BC, we believe that tiny amount of C₂H₄ may be also possibly formed on BC, but the yield is too low and below the detection limitation of the gas chromatography.

No observation of the infrared peak for H₂O peak at 1600 cm⁻¹ was caused by the measurement method for the *in-situ* infrared experiment. The signals of both adsorbed CO₂ and H₂O molecules were intentionally dislodged through background deduction to exclude interference information of the infrared peak of the reactants before light

irradiation, which could benefit for the study of the infrared peaks of the various intermediates after illumination. The similar measurement method for the *in-situ* infrared experiment has been widely reported, such as in *Nat. Energy* **2019**, *4*, 690; *Nano Energy* **2020**, *69*, 104421; *Joule* **2018**, *2*, 1004; *Chem* **2020**, *6*, 2335; *Angew. Chem. Int. Ed.* **2018**, *57*, 16447; *Angew. Chem. Int. Ed.* **2021**, *60*, 8705.

Figure 19. *In situ* FTIR spectra for the adsorption and activation of CO₂ on (a, b) VS-SAL₁₀, (c) SAL and (d) BC, respectively.

9. As to “*In the proposed research pathway for C₂H₄ formation, CO was considered as an important intermediate for the further C₂H₄ formation by the DFT calculation. Can it be confirmed by experiment, such as using CO as the starting reactants for the photocatalytic performance?*”

Using CO as the starting reactants has been conducted for the photocatalytic performance in this revision. The result reveals that considerable amount of C₂H₄ was

indeed detected (Figure S21). In addition, small amount of ethane (C_2H_6) and propylene (C_3H_6) were also produced. It indicates that use of CO as starting reactants may be further favorable for C-C, even C_2 -C coupling. The detail mechanism will be investigated in our later work.

Figure S21. Photocatalytic gasses evolution amounts as a function of light irradiation times of $V_S\text{-SAL}_{10}$ using CO as starting reactants.

Reviewer #3:

This is an interesting and potentially quite important manuscript. The concept that one can design surface sites that direct the coupling of carbons in CO_2 to form C_{2+} species is an important advance in CO_2 chemistry and heterogeneous catalysis in general. Overall the science and conclusions presented are of good quality.....

We much appreciate the positive comment.

1. As to *“This paper needs to be edited by a native speaker. Beyond simple issues of grammar and word choice, there are several sentences in the manuscript that I cannot understand and I’m certain don’t mean to convey the meaning that the selected words suggest.”*

The language of the revision has been much improved to make the manuscript more understandable.

2. As to “*On the one hand, the authors indicate that their material is essentially one unit cell thick. But, they also repeatedly talk about surface vs. bulk states. How can one have bulk states if there is no bulk?*”

One unit cell of AgInP_2S_6 actually consists of a sulfur framework with the octahedral voids exposing at outside surface, and Ag, In and P–P triangular patterns which fill in the sulfur framework. In the present manuscript, the surface of the monolayer is thus considered the surface S atom layer, and the bulk to interior Ag, In, and P, as described in the crystalline models of SAL.

Figure Crystalline model showing the surface and bulk parts (Only for review).

3. As to “*Neither in the manuscript or the SI are sufficient details given to reproduce the electrochemistry that is reported.*”

◆ “*How is the electrode made? What material is the AgInP_2S_6 attaching to?*”

The sufficient detail of the electrode fabrication for the electrochemistry measurement has been provided in the revision. The catalyst ink was prepared by dispersing 10 mg of as-prepared catalysts in 1 mL of ethanol under sonication. Then, 50 μL of the ink was evenly spread onto a piece of pretreated FTO within a 1 cm^2 area and dried under room temperature. The AgInP_2S_6 was thus attached to FTO.

◆ “*How did the researchers assure (or test for) the formation of an ohmic back contact? What was the working area of the electrode?*”

The solid-state current–voltage (J–V) test curves exhibit Ohmic characteristics (Figure

S25), confirming the formation of an ohmic back contact between samples and FTO.

The working area of the electrode is as large as 1cm^2 .

Figure S25. Solid-state current-voltage curves of BC, SAL and V_S -SAL₁₀.

◆ *“What scan rate(s) was used in Figure S9? What reference electrode and counterelectrode were employed? What is the composition of the electrolyte?”*

The scan rate was 5 mV s^{-1} . The reference electrode was the saturated Ag/AgCl electrode, and a Pt foil was employed as the counter electrode. The $0.5\text{ M Na}_2\text{SO}_4$ aqueous solution was used as the electrolyte.

◆ *“Figures S9a-c are referred to as Mott-Shottky plots. However, that should be a plot of $1/C^2$ vs. potential. The units on the vertical axis are not correct for MS plots. It would also help if the intercepting potentials shown on the figures were labeled “Flatband Potential”.*

Thank for kindly reminding. The units on the vertical axis have been corrected, and the Flatband Potential” was also labeled in the revision.

◆“Figure S11 is called “Gas chromatogram and mass spectra...”. But, there is only *ms* data. No *gc* data is provided. Figure S12 is referred to in the S11 figure caption. But, the discussion does not go with the S12 figure that is provided. It looks like a figure might be missing.”

Thank for kindly reminding. The gas chromatogram traces were supplemented as (a) and (b).

Figure S15. Gas chromatogram traces (a, b) and mass spectra (c, d) of $^{13}C_2H_4$ and $^{18}O_2$ produced over V_S -SAL₁₀. $^{13}CO_2$ and $H_2^{18}O$ were used.

REVIEWERS' COMMENTS

Reviewer #2 (Remarks to the Author):

In the revised manuscript, the most of my questions have been reluctantly answered. However, there is still some little problem needed to be clear or revised before publication.

(1) In my question (6) about the stability of photocatalyst, the authors have confirmed the instability of photocatalyst for 12 hour reaction and suggests that the post washing treatment can recover the catalytic activity. I hope the authors can provide the actual activity with 12 hour reaction for reader reference and the results of post washing treatment.

(2) The authors has misunderstood my question (7). I mean that besides the TPD peak, the mass spectra of corresponding TPD peak of CO should be provided simultaneously.

(3) The y-axis of infrared spectrum (Figure S19) in the revised manuscript is absorbance or transmittance or others, should be given.

Reviewer #3 (Remarks to the Author):

The authors have responded to all of my concerns and provided the important missing details. I believe the paper is now publishable in Nature Comm.

Our responses to the referees' comments are as follows;

Reviewer #2:

(1) In my question (6) about the stability of photocatalyst, the authors have confirmed the instability of photocatalyst for 12 hour reaction and suggests that the post washing treatment can recover the catalytic activity. I hope the authors can provide the actual activity with 12 hour reaction for reader reference and the results of post washing treatment.

The actual activity with 12-hour reaction and the results of post washing treatment were provided in this revision. After 12 h light irradiation, increase tendency of the generation of the hydrocarbon products slowed down (Figure S16a). It may be assigned to the potential carbon deposition as intermediates covering the active sites of the photocatalyst during photoreduction process. The problem may be resolved through post washing treatment to recover the catalytic activity to a certain extent, as shown in Figure S16b, although the deposited carbon can not be absolutely removed in the present case. Anyway, more effective methods to completely re-expose activity sites for recycle utilization of the photocatalyst is still explored currently.

Figure S16. Photocatalytic CO₂ activity of (a) fresh Vs-SAL and (b) post washing treated-Vs-SAL after 12-hour photocatalysis reaction.

(2) The authors has misunderstood my question (7). I mean that besides the TPD peak, the mass spectra of corresponding TPD peak of CO should be provided simultaneously.

The mass spectra of corresponding TPD peaks of CO were provided in the revision.

Figure S19. (a) CO TPD spectra of SAL and V_S-SAL₁₀. (b, c) Mass spectra of corresponding TPD peaks of CO desorbed from SAL and V_S-SAL₁₀, respectively.

(3) The y-axis of infrared spectrum (Figure S19) in the revised manuscript is absorbance or transmittance or others, should be given.

The y-axis of infrared spectrum (Figure S20) was given as transmittance.

Reviewer #3:

The authors have responded to all of my concerns and provided the important missing details. I believe the paper is now publishable in Nature Comm.

We appreciate very much for the reviewer's positive comment on our manuscript to be publishable.